# Morphine self-administration is inhibited by the antioxidant N-acetylcysteine and the anti-inflammatory ibudilast; an effect enhanced by their co-administration

**María Elena Quintanilla**[1,2☯], **Paola Morales**[1,2,3☯], **Daniela Santapau**[4], **Javiera Gallardo**[4], **Rocío Rebolledo**[1], **Gabriel Riveras**[1], **Tirso Acuña**[1], **Mario Herrera-Marschitz**[1], **Yedy Israel**[1,2], **Fernando Ezquer** [ID][4,5]*

1 Molecular and Clinical Pharmacology Program, Institute of Biomedical Sciences, Faculty of Medicine, Universidad de Chile, Santiago, Chile, 2 Specialized Center for the Prevention of Substance use and the Treatment of Addictions (CESA), Faculty of Medicine, University of Chile, Santiago, Chile, 3 Department of Neuroscience, Faculty of Medicine, Universidad de Chile, Santiago, Chile, 4 Center for Regenerative Medicine, Faculty of Medicine Clínica Alemana-Universidad del Desarrollo, Santiago, Chile, 5 Research Center for the Development of Novel Therapeutics Alternatives for Alcohol Use Disorders, Santiago, Chile

☯ These authors contributed equally to this work.
* eezquer@udd.cl

## Abstract

### Background

The treatment of opioid addiction mainly involves the medical administration of methadone or other opioids, aimed at gradually reducing dependence and, consequently, the need for illicit opioid procurement. Thus, initiating opioid maintenance therapy with a lower level of dependence would be advantageous. There is compelling evidence indicating that opioids induce brain oxidative stress and associated glial activation, resulting in the dysregulation of glutamatergic homeostasis, which perpetuates drug intake. The present study aimed to determine whether inhibiting oxidative stress and/or neuroinflammation reduces morphine self-administration in an animal model of opioid dependence.

### Methods

Morphine dependence, assessed as voluntary morphine self-administration, was evaluated in Wistar-derived UChB rats. Following an extended period of morphine self-administration, animals were administered either the antioxidant N-acetylcysteine (NAC; 40 mg/kg/day), the anti-inflammatory ibudilast (7.5 mg/kg/day) or the combination of both agents. Oxidative stress and neuroinflammation were evaluated in the hippocampus, a region involved in drug recall that feeds into the nucleus accumbens, where the levels of the glutamate transporters GLT-1 and xCT were further assessed.

**Data Availability Statement:** All data supporting this study are included within the article and supporting material.

**Funding:** This work was supported by Agencia Nacional de Investigación y Desarrollo (ANID) FONDECYT 1240162 and ACT210012 grants to Fernando Ezquer, and FONDECYT 1231443 to Mario Herrera-Marschitz. The funders had no role in study design, data collection and analysis, decision to publish, or preparation of the manuscript.

## Results

Daily administration of either NAC or ibudilast led to a mild reduction in voluntary morphine intake, while the co-administration of both therapeutic agents resulted in a marked inhibition (-57%) of morphine self-administration. The administration of NAC or ibudilast markedly reduced both the oxidative stress induced by chronic morphine intake and the activation of microglia and astrocytes in the hippocampus. However, only the combined administration of NAC + ibudilast was able to restore the normal levels of the glutamate transporter GLT-1 in the nucleus accumbens.

## Conclusion

Separate or joint administration of an antioxidant and anti-inflammatory agent reduced voluntary opioid intake, which could have translational value for the treatment of opioid use disorders, particularly in settings where the continued maintenance of oral opioids is a therapeutic option.

## 1. Introduction

Numerous studies have shown that repeated administration of morphine, the most widely used opioid for treating acute and chronic pain, induces cerebral oxidative stress [1–3] and activates proinflammatory responses in microglia and astrocytes—hallmarks of neuroinflammation—in brain regions associated with morphine dependence, including the hippocampus; ventral tegmental area, nucleus accumbens and prefrontal cortex [4–8]. With respect to the mechanism(s) through which morphine induces oxidative stress, previous studies have shown that morphine-induced activation of μ-opioid receptors inhibits excitatory amino acid transporter type 3 (EAAT3)-mediated cysteine transport into neurons, resulting in decreased levels of glutathione (GSH) and elevated oxidative stress [9]. In addition, morphine can induce oxidative stress via the stimulation of the mesolimbic dopaminergic system, as released dopamine is oxidized by monoamine oxidase, generating hydrogen peroxide and hydroxyl radicals [10]. Regarding the mechanism by which morphine generates neuroinflammation, it involves the activation of the pattern recognition receptor for foreign molecules Toll-like receptor 4 (TLR4) on microglia and astrocytes [11–13]. This activation, stimulates the nuclear translocation of the transcription factor NF-κβ [14], leading to the production of pro-inflammatory mediators such as tumor necrosis factor-alpha (TNF-α), interleukin-1β (IL-1β), and IL-6 [15]. In turn, TNF-α increases the generation of mitochondrial superoxide ions [16] and activates NADPH oxidase, generating hydrogen peroxide [17], resulting in oxidative stress. Compelling evidence suggests that once established, oxidative stress and neuroinflammation constitute a self-perpetuating cycle [16, 18, 19], wherein oxidative stress triggers neuroinflammation, which in turn, leads to or further maintains oxidative stress.

The finding that induction of both oxidative stress and neuroinflammation are observed upon morphine self-administration suggests that these events contribute to the reinforcing properties of opioids. Indeed, the administration of a plasmid encoding for the anti-inflammatory cytokine IL-10 into the nucleus accumbens reduced remifentanil self-administration [20]. Similarly, the antioxidant N-acetylcysteine has proven to be effective in preventing both the reinstatement of heroin-seeking behavior and the reinstatement of morphine-induced place conditioning (CPP) [21, 22]. Both oxidative stress and neuroinflammation directly contribute

to the dysregulation of glutamate homeostasis assessed by the levels of the glutamate transporters GLT-1 and xCT, crucially linked to drug-seeking behaviors and subsequent addiction [23].

In view of the aforementioned findings, toning down the degree of opioid dependence by clinically available antioxidant and anti-inflammatory agents could be valuable for the treatment of opioid addiction in humans, which often involves the medical administration of oral methadone or other opioids, aimed at gradually reducing dependence and thus the need for illicit opioid procurement [24, 25].

N-acetylcysteine is an antioxidant drug given its ability to elevate intracellular levels of L-cysteine and of glutathione (GSH), thereby providing reducing equivalents [26, 27]. Acting by a different mechanism, ibudilast is a drug that suppresses the proinflammatory response of glia [6] and exhibits phosphodiesterase activity [28]. Of interest, ibudilast also functions as a TLR4 antagonist, as supported by *in silico* analysis demonstrating that ibudilast docks to the same domain of the lipopolysaccharide (LPS) attachment [29]. In animal studies, ibudilast has demonstrated efficacy in decreasing self-administration of methamphetamine and cocaine [30, 31], as well as for reducing alcohol consumption in mice and rats [32]. In the context of opioid dependence, research has shown that ibudilast reduces morphine withdrawal syndrome precipitated by naloxone administration [6]. Recent studies carried out by Bates et al. (2024) [33] reported that N-acetylcysteine reduces naloxone-precipitated fentanyl withdrawal syndrome in rats. Nevertheless, no studies have been published as to whether N-acetylcysteine or ibudilast inhibit opioid self-administration in an animal model of opioid dependence.

Considering the above, in the present study, we hypothesized that in an animal model of well-established opioid dependence, the administration of (a) the antioxidant N-acetylcysteine (NAC) and (b) the anti-inflammatory ibudilast could effectively reduce chronic morphine intake. Furthermore, we hypothesized that a possible additional advantage might be observed when both drugs are co-administered.

## 2. Materials and methods

### 2.1 Induction of voluntary morphine consumption

Naïve female eight weeks-old rats from the UChB line, selectively bred for ninety generations for their high voluntary ethanol intake [34, 35], were used to induce voluntary morphine consumption as previously described [36]. All experiments were conducted with female rats. The rationale for choosing to work only with females was twofold: (i) we had previously demonstrated that female rats of this strain developed morphine dependence after achieving a stable morphine intake, as observed in the present study (15.6 ± 0.8 mg/kg/day) using the same induction method [36]; and (ii) female rats were selected for translational considerations, such as the urgent need to manage opioid dependence in pregnant women, who are typically treated with long-acting opioids (e.g. methadone). We have recently shown that methadone itself is a neurotoxin [37]. Nevertheless, the fact that we did not include male animals is a limitation of the study. Animals were maintained on a 12-hour light/dark cycle (lights were turned off at 7:00 p.m.) and were regularly fed a rodent diet containing soy protein and peanut meal (Cisternas, Santiago, Chile). The diet utilized is primarily composed of soy protein, avoiding fish proteins due to the presence of calcined bonemeal, which has been shown to contain cyanamide, an inhibitor of aldehyde dehydrogenase (ALDH2) [38]. Animal procedures were approved by the local Committee for Experiments with Laboratory Animals of the Faculty of Medicine at the University of Chile (CBA Protocol # 0994 FMUCH).

As previously reported, the severe morphine dependence rat model used here shows the full eight classical dependence signs triggered by naloxone administration [36]. For the induction of voluntary oral morphine consumption, animals (n = 24) were housed in individual cages

with food and water *ad libitum*, administering an intraperitoneal (i.p.) priming dose of 40 mg/kg of morphine hydrochloride trihydrate daily for 9 consecutive days to induce subsequent oral voluntary morphine preference. On day 10, the morphine injections were discontinued, and each cage was fitted with a second drinking tube (two bottle-choice paradigm) one containing tap water and the other morphine sulfate solution (6 mg/L). Subsequently, the concentration of the morphine sulfate solution was progressively increased every three days, reaching a maximum of 90 mg/L on day 30 [36]. The levels of morphine self-administration at different times prior to antioxidant or anti-inflammatory drug treatments are shown in **S1 Fig**.

## 2.2 Drugs and treatments

Morphine hydrochloride trihydrate (20 mg/mL, Sanderson Laboratory, Santiago, Chile) was used for the initial intraperitoneal administrations, injected at 40 mg/kg/day. For oral consumption, daily morphine solutions were prepared by dissolving morphine sulfate (20 mg/mL, Oramorph, Molteni Farmaceutici, Italy) in distilled water. The concentrations of the morphine sulfate solution (calculated as the base) for oral intake ranged from 6 to 90 mg/L (w/v). The oral route was chosen for morphine administration because it has been previously reported that this protocol induces opioid dependence (assessed by naloxone-precipitated withdrawal syndrome) in UChB rats [36]. Additionally, it has been reported that in humans over 90% of prescription opioid abusers report oral ingestion for nonmedical purposes [39, 40]. N-acetylcysteine (NAC), obtained from Sigma-Aldrich (St. Louis, MO), was prepared by dissolving it in saline and adjusting the pH to 7.2. Initially, animals were administered a high-loading dose of NAC (70 mg/kg/day intraperitoneally (i.p.) in a volume of 5.0 mL/kg for two consecutive days. This was followed by a maintenance dose of NAC of 40 mg/kg/day, in a volume of 5.0 mL/kg, i.p. for four days. We chose these loading and maintenance doses of NAC because we found them to be effective in reducing alcohol relapse previously [41]. In addition, the maintenance dose of NAC chosen (40 mg/kg per day) was similar to that used in humans for cocaine treatment [42]. Ibudilast (Sigma-Aldrich, St. Louis, MO) was dissolved in a solution of 35% polyethylene glycol in saline and administered at a dosage of 7.5 mg/kg, i.p., once daily for four consecutive days, as previously reported [5]. This dose of ibudilast was chosen because it effectively blocks drug-induced reinstatement of morphine conditioned place preference following morphine re-exposure [5]. When NAC and ibudilast were co-administered, each was dissolved separately, as described above, and NAC was administered immediately before ibudilast.

## 2.3 Effects of intraperitoneal administration of NAC or ibudilast, given separately, and in combination on chronic morphine intake

From day 30 to day 57, the 24 rats described above had free access to two bottles: one containing a 90 mg/L morphine sulfate solution and the other containing tap water. On day 51 of continuous (24 h/day) voluntary morphine consumption, the animals were divided into four groups, each comprising six rats that were administered by the intraperitoneal route as follows: **(a) Vehicle group**: Rats received a solution of 35% polyethylene glycol in saline (0.9% NaCl) at a volume of 5.0 mL/kg for six days; **(b) Ibudilast group**: Rats were administered saline (at a volume of 5.0 mL/kg) for the first two days, followed by ibudilast (7.5 mg/kg/day) at a volume of 2.5 mL/kg for the next four days; **(c) NAC group**: Rats received a dose of 70 mg/kg/day of NAC (at a volume of 5.0 mL/kg) for the initial two days, followed by a dose of 40 mg/kg/day of NAC (at a volume of 5.0 mL/kg) for the next four days; **(d) NAC + ibudilast group**: Rats received 70 mg/kg/day NAC (at a volume of 5.0 mL/kg) during the first two days, and over the next four days they first received a dose of 40 mg/day of NAC, followed immediately by a dose of ibudilast (7.5 mg/kg/day). A parallel group of 6 female rats, drinking water only was used as

a control. Daily morphine intake was recorded, expressed as milligrams of morphine sulfate consumed per kilogram of body weight per day. A timeline showing the experimental design is shown in **S2 Fig**.

Seventeen hours after the last administration of NAC and ibudilast and one hour after the last recorded morphine intake, rats that continued to have access to the morphine solution and water were anesthetized with a cocktail of 60 mg/kg ketamine HCl, 10 mg/kg xylazine HCl and 4 mg/kg acepromazine (administered intramuscularly in a volume of 1.9 mL/kg) [43], intracardially perfused with 100 mL of 0.1 M PBS (pH 7.4) and euthanized. The brain was rapidly removed to dissect the hippocampus, using one hemisphere for determination of oxidative stress and the hippocampus of the contralateral hemisphere for immunofluorescence analysis of the astrocyte marker glial fibrillary acidic protein (GFAP), and the microglial marker ionized calcium binding adaptor molecule 1 (Iba-1). Nucleus accumbens were also dissected out for determining GLT-1 and xCT levels.

## 2.4 Determination of hippocampal oxidative stress

Oxidative stress was assessed by determining the ratio between oxidized glutathione (GSSG) and reduced glutathione (GSH), as well as lipid peroxidation levels in one hippocampus. Meanwhile, to reduce the number of animals to be used, the contralateral hippocampus was utilized to evaluate astrocyte and microglia immunoreactivity (see below). The GSSG/GSH ratio was determined as previously reported [44]. Lipid peroxidation was quantified by malondialdehyde (MDA) as previously described [36], using a Lipid Peroxidation Assay kit (Sigma-Aldrich).

## 2.5 Determination of astrocyte and microglia immunoreactivity

After intracardially perfusion with 100 mL of 0.1 M PBS, one of brain hemisphere was dissected, fixed in 4% paraformaldehyde (Merck) and cryopreserved in 10% and 30% saccharose as previously reported [44]. Double labeling immunofluorescence against glial fibrillary acidic protein (GFAP), an intermediate filament protein present in astrocytes (Sigma-Aldrich G3893; 1:500 dilution), and against the microglial marker ionized binding protein1 (Iba1) (Wako 019–19741, 1:400 dilution) were evaluated in coronal cryosections (30 μm thick). Nuclei were counterstained with DAPI (4,6-diamino-2-phenylindole, Invitrogen, 0.02 M; 0.0125 mg/mL). Microphotographs of the *stratum radiatum* of the CA1 region of the hippocampus were assessed using an Olympus FV10i confocal microscope. The area analyzed for each stack was 0.04 mm$^2$ and for each case the Z-axis thickness was also recorded. The density of GFAP-positive astrocytes and the length size of primary astrocytic processes were quantified using a FIJI analysis software, as previously reported [44]. Reactive astrocytes typically undergo hypertrophy, which involves an increase cell density and increase in cell size by enlargement of their processes in length and in thickness [45]. The density of Iba-1 positive microglia and the length, size, and thickness of their primary processes were quantified in the hippocampus using FIJI analysis software, as previously reported [44]. Iba-1 positive microglia exhibiting shorter primary processes and larger cell bodies with a rounded or amoeboid phenotype, similar to that of macrophages and phagocytic structures, suggesting a phagocytic function [46, 47] were called "reactive" microglia, while those cells exhibiting long primary processes and small cell bodies were called "surveillance" microglia, as previously suggested [48].

## 2.6 Determination of glutamate transporter 1 (GLT-1) and glutamate-cystine exchanger (xCT) protein levels

Total proteins in samples of nucleus accumbens were extracted using a T-per lysis buffer (Thermo-Fisher) containing protease inhibitors. For Western Blots, 25 μg of proteins was

utilized to assess GLT-1 levels with a guinea pig anti-GLT-1 primary antibody (Millipore, AB1783, 1:500 dilution) and an IRDye 800CW donkey anti-guinea pig secondary antibody (Li-COR, 925–32411, 1:10,000 dilution); while xCT levels were detected with a rabbit anti-xCT primary antibody (Abcam, AB175186,1:500 dilution) and an IRDye 800CW donkey anti-rabbit secondary antibody (LI-COR, 926–32213, 1:10,000 dilution). The same membranes were probed for GAPDH reactivity as a loading control, using a rabbit anti-GAPDH primary antibody (Cell Signaling, 2118,1:1,000 dilution) and an IRDye 680CW goat anti-rabbit secondary antibody (Abcam, AB2167771:10,000 dilution). Detection of reactive bands were performed using the Odyssey Imagen System (Li-COR) and analyzed with Image Studio Lite 5.2 software as previously reported [36].

## 2.7 Statistical analyses

Data are expressed as means ± SEM. Statistical analyses were performed using GraphPad Prism v.9.2.0 software. The normal distribution of data for all experiments was evaluated using the Shapiro-Wilk test. For normally distributed data, one-way ANOVA (Figs 3, 4B, 5 and 6) or two-way ANOVA (Figs 1 and 2 and S1) was used followed by a Tukey or Dunnett *post hoc* test. Only data of one study (Fig 4A) did not exhibit a normal/Gaussian distribution; thus, in this case data were analyzed using a non-parametric Kruskal-Wallis test followed Dunn's post hoc test. When only two groups were compared, statistical significance was determined by Student's t-test. A level of $p < 0.05$ was considered for statistical significance. For greater fluidity in reading the RESULTS section, the complete statistical analysis is provided in the corresponding Figure legend.

## 3. Results

### 3.1 Morphine self-administration

**Fig 1** illustrates that under free-choice concurrent access to a morphine solution (90 mg/L) and water, rats exhibited an average consumption of 14.3 ± 0.2 mg of morphine/kg body weight/day (mean ± SEM) and displayed a high preference for morphine over water (more than 80%). The morphine intake during the baseline period was not significantly different among the four groups of rats (two-way ANOVA). Administration of a loading dose of NAC (70 mg/kg/day; i.p.) for two days (days 50 and 51 of morphine intake) to rats in both the NAC group (green squares) and the NAC + ibudilast group (red squares) resulted in a significant reduction in chronic morphine intake compared to the other two groups treated with vehicle (the control group [black squares] and the group assigned to receive ibudilast [blue squares]) (***p<0.001 for the NAC group and **p<0.01 for the NAC + ibudilast group, respectively, two-way ANOVA followed by Dunnett's post hoc test). The rationale for administering loading dose was to rapidly attain elevated plasma levels of NAC, facilitating the prompt neutralization of free radicals generated by prolonged morphine consumption [49]. Subsequently, during days 53, 54, 55, and 56, the rats in all four groups were treated with either: (i) vehicle (filled black squares), (ii) NAC at a dose of 40 mg/kg/day (filled green squares), (iii) ibudilast at a dose of 7.5 mg/kg/day (filled blue squares) or (iv) NAC (40 mg/kg/day) plus ibudilast (7.5 mg/kg/day) (filled red squares). The rationale for reduce the NAC dose during this period compared to that administered on days 51 and 52 was intended to avoid counteracting a possible plateau inhibitory effect of NAC when combined with ibudilast in the NAC + ibudilast group. A two-way ANOVA, followed by Tukey's post-hoc test, revealed that morphine intake by the four groups of rats during days 53, 54, 55, and 56 was significantly reduced by both NAC and ibudilast treatments, as well as by the combination of NAC + ibudilast, compared to the morphine + vehicle group.

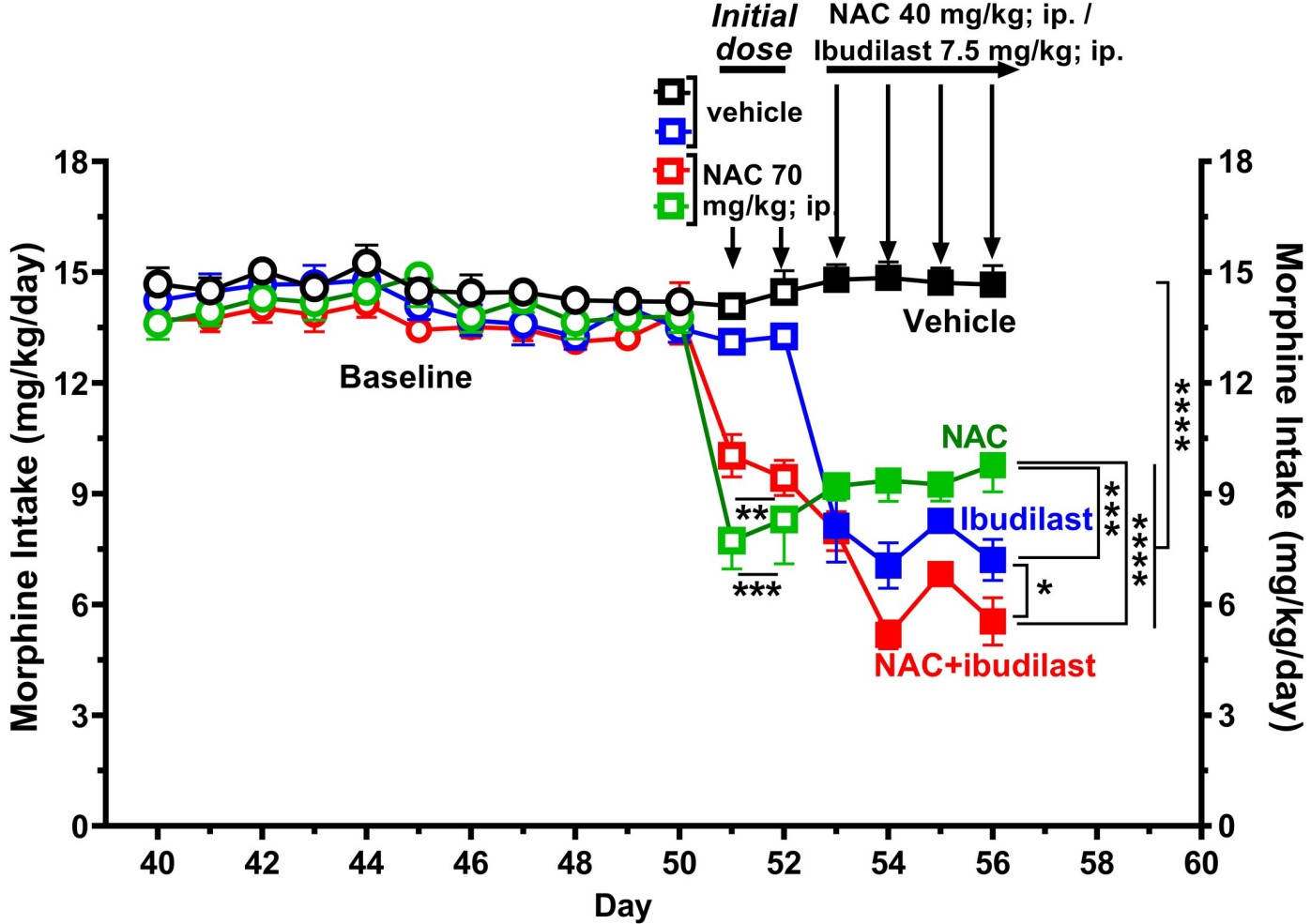

**Fig 1. The co-administration of NAC (40 mg/kg/day, i.p.) and ibudilast (7.5 mg/kg/day, i.p.) leads to a greater reduction in chronic morphine intake compared to separate administration of both drugs.** Figure shows morphine consumption expressed as mg of morphine consumed per kilogram of body weight per day (mean ± SEM) of rats subjected to free choice between a 90 mg/L morphine sulfate solution and water for fifty-six consecutive days. Shapiro-Wilk tests conducted for all morphine intake data indicated that data from all groups passed the normality test (alpha = 0.05), implying that the dataset follows a normal distribution. A two-way ANOVA (groups × day) of the baseline morphine intake data revealed no significant differences between the four baseline groups [$F_{baseline(3,20)}$ = 1.454, p: N.S.]. A two-way ANOVA of the morphine intake data from all four groups on days 50 and 51 of treatment, during the administration of a loading dose of NAC (70 mg/kg/day) to rats in both the NAC and NAC + ibudilast groups, revealed a significant effect of the NAC loading dose in both groups compared to the other two vehicle-treated groups [$F_{treatment}(3,20)$ = 27.14, p<0.0001] but not of day. Dunnett post-hoc test indicated that the administration of NAC induced a significant reduction of morphine intake in both the NAC (***p< 0.001) and the NAC + ibudilast group (**p<0.01) versus that of groups treated with vehicle. A two-way analysis of variance (ANOVA) of morphine intake data obtained during days 53, 54, 55 and 56 of the four groups indicates a significant effect of treatment [$F_{treatment(3,20)}$ = 24.10, p<0.0001], day [$F_{day(1914,38.28)}$ = 5.822, p<0.0068] and a significant treatment × day interaction [$F_{interaction(9,60)}$ = 4.035, p<0.0004] versus that of the vehicle treated group (control). Tukey's post hoc analysis revealed that the administration of: i) NAC + ibudilast, ii) NAC or iii) ibudilast induced a significant reduction in chronic morphine intake (****p<0.0001; n = 6 rats/group) compared to vehicle-treated group (control). In addition, the reduction induced by the co-administration of NAC + ibudilast was significantly greater than that induced by NAC (****p<0.0001) and ibudilast (*p<0.05) administered separately, while the reduction of morphine intake induced by ibudilast was significantly higher than that induced by NAC (***p<0.001).

It's worth noting that the co-administration of NAC + ibudilast resulted in a significantly greater inhibition of morphine intake (57% reduction) compared to the inhibition observed in animals administered only NAC (36% reduction, p<0.0001, two-way ANOVA followed by Tukey's post-hoc test) or only ibudilast (48% reduction, p<0.05, two-way ANOVA followed by Tukey's post-hoc test).

In relation to total water homeostasis, **Fig 2A** illustrates that the basal water consumption of the four groups of rats did not differ significantly (two-way ANOVA). The administration of a loading dose of NAC (70 mg/kg/day; i.p.) over two days (on days 50 and 51) to both the NAC group (green squares) and the NAC + ibudilast group (red squares) resulted in a significant increase in water intake compared to the vehicle-treated groups. The administration of NAC, ibudilast, or the combination of NAC + ibudilast (all of which reduced voluntary morphine solution intake) induced a significant increase in water intake on days 53, 54, 55, and 56 (p<0.0001, two-way ANOVA followed by Tukey's post-hoc test). The observation that this increase in water intake coincided with a reduction in morphine intake suggests that increased water intake is a compensatory mechanism. Indeed, as shown in **Fig 2B**, total fluid intake was not affected by NAC, ibudilast, or NAC + ibudilast treatments (p = 0.81 N.S., two-way ANOVA, indicating that reduction in morphine intake was not due to nonspecific discomfort. Furthermore, normal body weight gain was not significantly affected by the treatments (p = 0.92 N.S., two-way ANOVA) (**Fig 2B**).

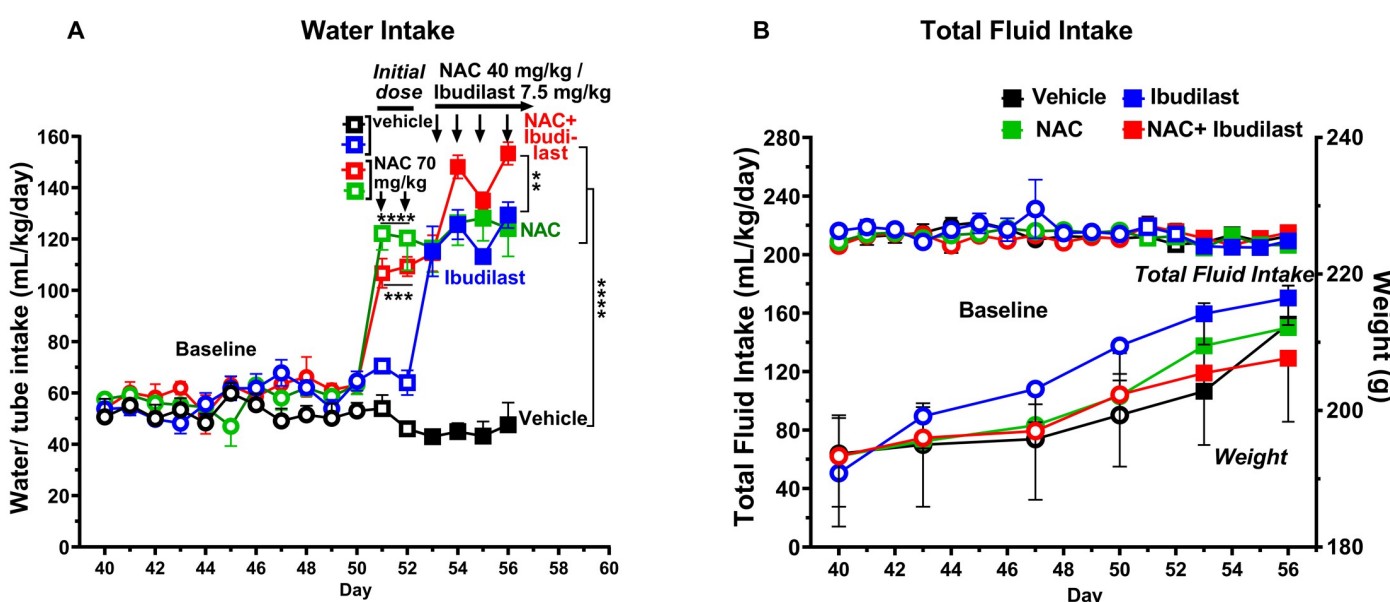

**Fig 2. The administration of NAC (40 mg/kg per day, i.p.) and ibudilast (7.5 mg/kg per day, i.p.) alone, as well as their co-administration for four days, significantly increased water intake in rats that had been consuming morphine for fifty-six days, without affecting total fluid intake or body weight.** Shapiro-Wilk tests conducted for all morphine intake data and weight data indicated that data from all groups passed the normality test (alpha = 0.05), implying that the dataset follows a normal distribution. Fig 2A shows water consumption expressed as mL of water consumed per kilogram of body weight per day (mean ± SEM) of rats subjected to free choice between a 90 mg/L morphine sulfate solution and water for fifty-six consecutive days. A two-way ANOVA (groups × day) of the baseline water intake data revealed no significant differences between the four baseline groups [$F_{baseline(3,20)}$ = 1.754, p: 01884. N.S.]. A two-way ANOVA (treatment × day) of water intake on both days (day 50 and day 51) of treatment with a loading NAC dose of 70 mg/kg/day to rats in both the NAC and NAC + ibudilast groups (and administration of vehicle to the other two groups) revealed significant effect of the loading dose of NAC compared with groups treated with vehicle [$F_{treatment(3,20)}$ = 41.37, p<0.0001] but not of day. Dunnett post-hoc test revealed that the administration of NAC induced a higher increase of water intake in both the NAC (****p<0.0001) and the NAC + ibudilast group (***p<0.001) versus that of groups treated with vehicle. A two-way ANOVA (treatment × day) of water intake data obtained during days 53, 54, 55, and 56 of the four groups revealed significant effect of treatment [$F_{treatment(3,20)}$ = 61.65, p<0.0001], day [$F_{day(1.866,37.31)}$ = 9.021, p<0.001] and a significant treatment × day interaction [$F_{interaction(9,60)}$ = 2.533, p<0.05] vs that of the vehicle treated group (control). Tukey's post hoc analysis revealed that administration of: i) NAC + ibudilast, ii) NAC or iii) ibudilast induced a significant increase in water intake (****p< 0.0001; n: 6 rats/group) compared to the vehicle-treated group. In addition, the increases induced by the coadministration of NAC + ibudilast was significantly greater than that induced by ibudilast (**p<0.01). Fig 2B (top) shows total fluid intake expressed as mL consumed per kilogram of body weight per day (mean ± SEM) and weight (g) (bottom) of rats subjected to free choice between a 90 mg/L morphine sulfate solution and water for fifty-six consecutive days. Shapiro-Wilk tests performed on data from either the total fluid intake groups or the body weight groups indicate that all datasets met the criteria for normality (alpha = 0.05), implying that both dataset groups follow a normal distribution. Two-way ANOVA (treatment × day) of all data shown that the administration of NAC, ibudilast, or the combination of NAC + ibudilast did not affect total fluid intake [$F_{treatment(3,20)}$ = 0.3134, p: 0.81, N.S.] or body weight [$F_{treatment(3,20)}$ = 0.1611, p: 0.92, N.S.].

### 3.2 Hippocampal oxidative stress

Oxidative stress was evaluated by determining the ratio of oxidized to reduced glutathione (GSSG/GSH) and by measuring the levels of lipid peroxidation, quantifying malondialdehyde (MDA) in the hippocampus. Both parameters are highly sensitive indicators of the brain's cellular redox state [50]. As showed in **Fig 3A and 3B**, chronic morphine consumption led to a three-fold increase in the GSSG/GSH ratio and a two-fold increase in MDA levels compared to the control group, consuming water only (p<0.01, one-way ANOVA followed by Tukey's post-hoc test). Morphine-induced increase in oxidative stress was significantly reduced by the treatment, with either (a) NAC; (b) ibudilast; or (c) co-administration of both drugs, assessed by the GSSG/GSH ratio (p<0.01, one-way ANOVA followed by Tukey's post-hoc test) or by MDA levels (p<0.01, one-way ANOVA followed by Tukey's post-hoc test).

### 3.3 Evaluation of morphine-induced neuroinflammation

Glial activation was evaluated as previously described [36, 41, 47, 48], determining astrocyte and microglial activation (cell density, length of cellular process, and phagocytic microglial structures) in the hippocampus after 56 days of morphine consumption and treatments.

**(a) Hippocampal Astrocytes.** **Fig 4** displays microphotographs from *Stratum radiatum* of the CA1 hippocampal region showing GFAP-positive astrocytes immunoreactivity. **Fig 4A and 4B** reveal that rats consuming morphine for 56 days exhibited a significant increase in the length size of primary astrocytic processes (p<0.0001, one-way ANOVA followed by Tukey's post-hoc test) compared to water-drinking animals. The administration of NAC, ibudilast, or

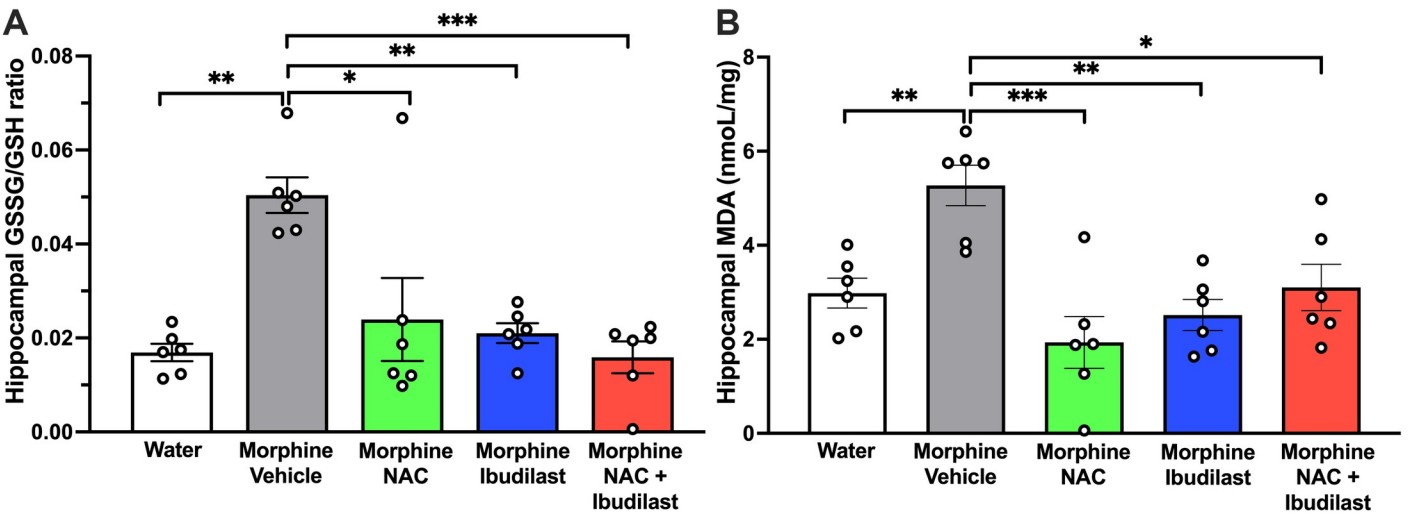

**Fig 3. The significant increase in oxidative stress observed in the hippocampus of rats that had been consuming morphine for fifty-six days was effectively mitigated by the separate administration of NAC (40 mg/kg/day, i.p.) and ibudilast (7.5 mg/kg/day, i.p.), as well as by their co-administration for four days.** Shapiro-Wilk tests performed on data from either the groups of GSSG/GSH ratio data (A) or the groups of MDA data (B) indicate that both datasets met the criteria for normality (alpha = 0.05), implying that the data for the GSSG/GSH ratio and the data for MDA levels follow a normal distribution. **(A)** Oxidative stress levels determined by measuring the ratio between oxidized/reduced glutathione (GSSG/GSH) (mean ± SEM) in the hippocampus. One-way ANOVA indicates significant effect of treatment [$F_{treatment(4,25)}$ = 6.563, p<0.001, n = 6 rats per group] compared to the control group treated with vehicle. Tukey's post hoc test revealed that vehicle-treated morphine-drinking rats showed a three-fold increase in the GSSG/GSH ratio (**p<0.01) compared to rats that drank only water. Administration of NAC (*p<0.05), ibudilast (**p<0.01) and NAC + ibudilast (***p<0.001) significantly reduced the increase in GSSG/GSH ratio compared to the control group treated with vehicle. **(B)** Oxidative stress levels determined by measuring the malondialdehyde (MDA) levels (mean ± SEM) in the hippocampus. One-way ANOVA indicates significant effect of treatment [$F_{treatment(4,25)}$ = 8.53, p<0.001, n = 6 rats per group] compared to the control group treated with vehicle. Tukey's post hoc test revealed that vehicle-treated morphine-drinking rats showed a two-fold increase in the MDA levels (**p<0.01) compared with rats drinking only water. Administration of NAC (***p<0.0001), ibudilast (**p<0.01) and NAC + ibudilast (*p<0.05) significantly reduced the increases in MDA levels compared to the control group treated with vehicle.

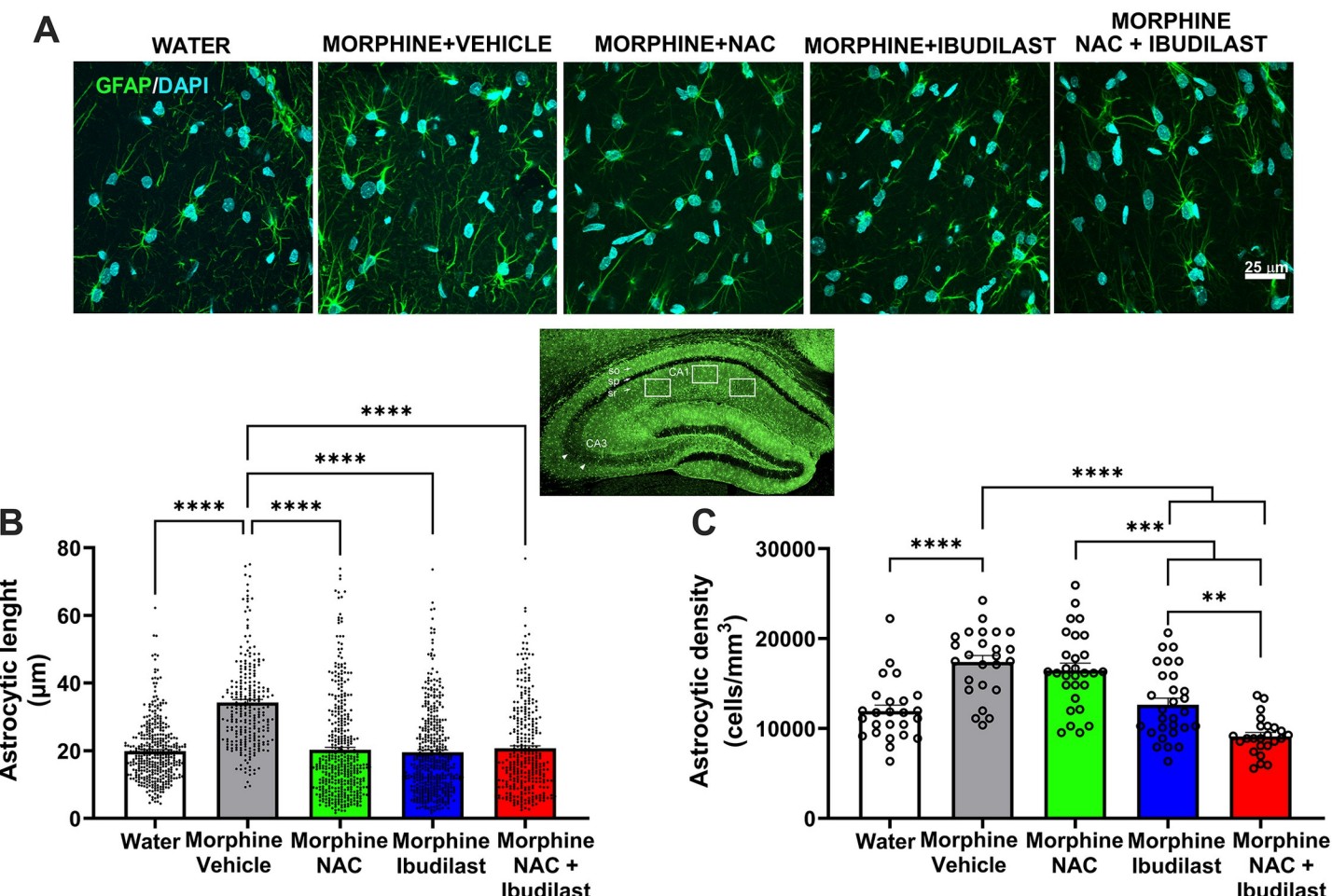

**Fig 4. The significant increase in astrocytic density and astrocyte activation observed in the hippocampus of rats consuming morphine for fifty-six days was reduced by the administration of NAC, ibudilast, and the combined administration of both drugs (NAC + ibudilast). (A)** Representative confocal microphotographs of GFAP immunoreactivity (green top) in hippocampal astrocytes. Nuclei were counterstained with DAPI (blue, nuclear marker), scale bar 25 μm. **(B)** Quantification of astrocyte density and **(C)** Length of primary astrocytic processes showing individual values. The inset displays a representative hippocampal microphotograph, indicating the area analyzed in *stratum radiatum (sr)* from CA1 region, depicted by white boxes. Shapiro-Wilk tests performed on data from either the groups of astrocytic length data (A) or the groups of astrocytic density data (B) indicate that only astrocytic density datasets met the criteria for normality (alpha = 0.05), while astrocytic length data did not pass normality tests, implying that data for the astrocytic density follow a normal distribution and length size of astrocytic processes data do not exhibit a normal/Gaussian distribution, thereby they were analyzed using the non-parametric Kruskal-Wallis test. (A) Kruskal-Wallis analysis of all astrocytic length data indicates a significant difference between groups (Kruskal-Wallis statistic: 181.7, ****$p < 0.0001$). Dunn's post-hoc test indicate that the morphine drinking group exhibits a higher length size of astrocytic process compared to water drinking rats (****$p < 0.0001$). The administration of NAC (***$p < 0.0001$), ibudilast (****$p < 0.0001$) and NAC + ibudilast (****$p < 0.0001$) fully normalized the morphine-induced increases in the length size of astrocytic process compared to the morphine drinking group treated with vehicle. (B) One-way ANOVA followed by Tukey's post-hoc test revealed that chronic morphine intake (56 days) by rats treated with vehicle (morphine + vehicle) led to an increase in astrocytic density [$F_{treatment(4,274)} = 26.0$, ****$p < 0.0001$], and length of astrocytic processes. At the administered doses, ibudilast, but not NAC, reduced the chronic morphine-induced increase in astrocytic density (****$p < 0.0001$). This inhibition was synergistically increased by the co-administration of NAC + ibudilast when compared with either the NAC (***$p < 0.001$) or ibudilast (**$p < 0.01$) groups.

the co-administration of NAC + ibudilast fully reversed the increase in the length size of the astrocytic processes ($p < 0.0001$, one-way ANOVA followed by Tukey's post-hoc test).

**Fig 4A and 4C** show that rats voluntarily consuming morphine exhibited a higher astrocyte density (number of GFAP$^+$ cells per mm$^3$) in the hippocampus compared to water-drinking animals ($p < 0.0001$; one-way ANOVA followed by Tukey's post-hoc test). Unexpectedly, only ibudilast treatment, but not NAC treatment, significantly attenuated the increase of morphine-induced density in GFAP$^+$ astrocytes, whether administered separately ($p < 0.0001$) (**Fig 4A and 4C**) or in

combination with NAC (p<0.0001), compared to the morphine vehicle group. Importantly, the co-administration of NAC + ibudilast led to a reduction in the morphine-induced increased density of GFAP$^+$ astrocytes, that was significantly greater than the reduction induced by the separate administration of NAC (p<0.001) or ibudilast (p<0.01) (**Fig 4C**).

**(b) Hippocampal Microglia.** Fig 5A, displays microphotographs from *Stratum radiatum* of the CA1 hippocampal region showing Iba-1 positive microglial immunoreactivity (in red, highlighted by arrows) and DAPI (nuclear marker in blue) from the five experimental groups. Panels A and B reveal that, as expected, the vehicle-treated morphine group exhibited a higher microglia density compared to the water-drinking control group (p<0.05). However, this morphine-induced increase in microglia density was not significantly decreased by the treatment, either with NAC, ibudilast, or their combination, suggesting that a more potent treatment is necessary to normalize microglia density. Since alterations in microglial morphology may occur in the absence of overall changes in tissue density, a finer diagnostic analysis of neuroinflammation was conducted by assessing the effect of chronic morphine self-administration on the ratio of reactive to surveillance hippocampal microglia and the density of phagocytic structures within the microglia, as previously reported [46, 47, 51]. As shown in Fig 5A and 5E, while the hippocampus of water-drinking rats (control) was enriched with highly ramified cell phenotypes (evidenced by a low ratio of reactive/surveillance cells), morphine-drinking rats contained a higher proportion of cells with shorter primary processes (evidenced by a high ratio of reactive/surveillance microglia) compared to animals that only consumed water (p<0.0001). These findings, which show that morphine induces both an increase in microglial density and a shift toward rounder, less ramified microglia, suggest that morphine has induced a homeostatic change toward a reactive microglial state. Importantly, the morphine-induced increase in the ratio of reactive to surveillance cells was significantly reduced by the administration of either NAC (p<0.0001), ibudilast (p<0.001), or the combination of both treatments (p<0.001), compared to the pattern observed in the morphine group treated with the vehicle (Fig 5A and 5E).

Additionally, the vehicle-treated morphine group exhibited a significantly higher density of phagocytic microglial structures (p<0.01, **Fig 5A and 5F**) compared to rats drinking only water. Morphine-induced increase in the density of phagocytic microglial structures was significantly reduced by NAC (p<0.001) or ibudilast (p<0.001) administration separately (**Fig 5F**), but unexpectedly not by the combination of both drugs (p = 0.34) compared to morphine-drinking rats treated with vehicle. The lack of efficacy in reducing morphine-induced microglial phagocytosis activation with the combined administration of NAC + ibudilast could be attributed to the potential beneficial role of phagocytosis in the removal of dead and dying neurons, as well as neuronal debris and reduction of neuroinflammation [52–54].

## 3.4 Glutamate homeostasis

The downregulation of glutamate re-uptake and the reduction in the expression of the glutamate transporter GLT-1, but not of the glutamate antiporter xCT in the nucleus accumbens, following extinction from heroin self-administration in rats has been suggested as a critical mechanism of opioid addiction [55]. To determine whether chronic morphine consumption is associated with a reduction in GLT-1 and/or xCT, we evaluated the glutamate transporters levels in the nucleus accumbens of rats that voluntarily drank morphine for 56 days, examining the effects of either NAC or ibudilast treatment on GLT-1 and xCT protein levels. It was found that morphine-drinking rats exhibited a significant reduction (~40%) in glutamate transporter GLT-1 protein levels (**p<0.001, **Figs 6A and 6B** and **S3**), but no changes in xCT protein levels (**Figs 6C and 6D** and **S3**). Furthermore, neither separate treatment with NAC nor with ibudilast reversed the reduction in GLT-1 levels induced by morphine (**Fig 6A and 6B**). However,

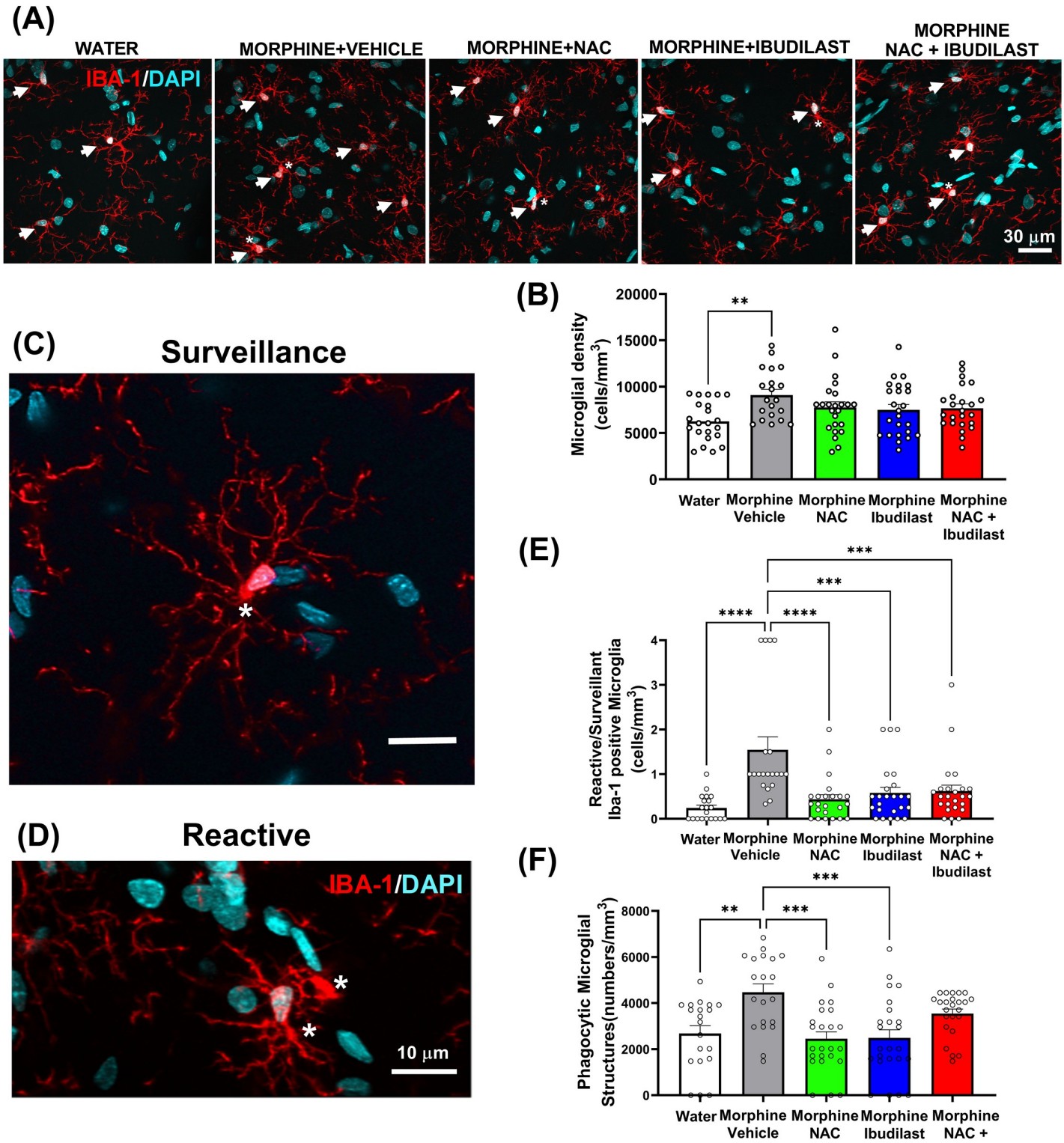

**Fig 5. The morphine-induced increase of reactive/surveillant microglial ratio, microglial cell density, and microglial phagocytic structures was fully normalized by the administration of NAC, ibudilast, and the combined administration of both drugs (NAC + ibudilast).** The figure shows representative confocal photomicrographs of immunofluorescence of microglia with immunoreactivity of Iba-1 (in red, indicated with arrows) and DAPI (blue nuclear marker). **(A)** Represents the five experimental

groups, scale bar 30 μm; **(C)** Represents a surveillance phenotype of anti-inflammatory microglial cell, scale bar 10 μm; and **(D)** Represents a reactive phenotype of pro-inflammatory microglial cell, denoting phagocytosed bead-like structures (asterisks), suggesting the presence of phagocytosed material, scale bar 10 μm. Data is presented as means ± SEM, showing individuals values. Shapiro-Wilk tests conducted for the groups of microglial density data (B), reactive/surveillant microglia ratio data (E), or the phagocytic microglial structures density data (F) indicated that all datasets met the criteria for normality (alpha = 0.05), implying that all data groups of microglial density, reactive/surveillant microglia ratio, as well as phagocytic microglial structures density follow a normal distribution. **(B)** One-way ANOVA of microglial density data revealed an increase of microglial density in morphine drinking rats treated with vehicle compared to rats drinking only water [One-way ANOVA followed by Tukey's post-hoc test, ($F_{\text{treatment}(4,109)}$ = 3.088, *p<0.05)]. The administration of NAC, ibudilast or NAC + ibudilast treatments did not reduce microglial density. **(E)** One way ANOVA of reactive/surveillant microglia ratio revealed that chronic morphine intake induces an increase of the reactive/surveillant ratio compared to rats that only drank water [One-way ANOVA followed by Tukey's post-hoc test ($F_{\text{treatment}(4,106)}$ = 9.482,****p<0.0001)]. The morphine-induced increase in reactive/surveillant ratio was reduced with the administration of NAC (****p<0.0001), ibudilast (***p<0.001), and the combined administration of both NAC and ibudilast (***p<0.001). **(F)** Analisis of phagocytic microglial structures indicated that chronic morphine intake induces an increased phagocytic microglial structures density compared to rats that only drank water [One way ANOVA followed by Tukey's post-hoc ($F_{\text{treatment}(4,107)}$ = 7.471, ****p<0.0001)]. The morphine-induced increase in phagocytic microglial structures density was significantly reduced when NAC or ibudilast were administered separately (Tukey post hoc ***p<0.001).

when NAC and ibudilast were co-administered, they completely normalized the reduction of GLT-1 levels induced by morphine in the nucleus accumbens (p<0.05, one-way ANOVA followed by Tukey post-hoc test) (**Fig 6A and 6B**).

## 4. Discussion

The data showed that Wistar-derived UChB rats voluntarily consumed morphine at an average of 14.30 ± 0.29 mg/kg/day (mean ± SEM) over a 4-week period. This intake was previously

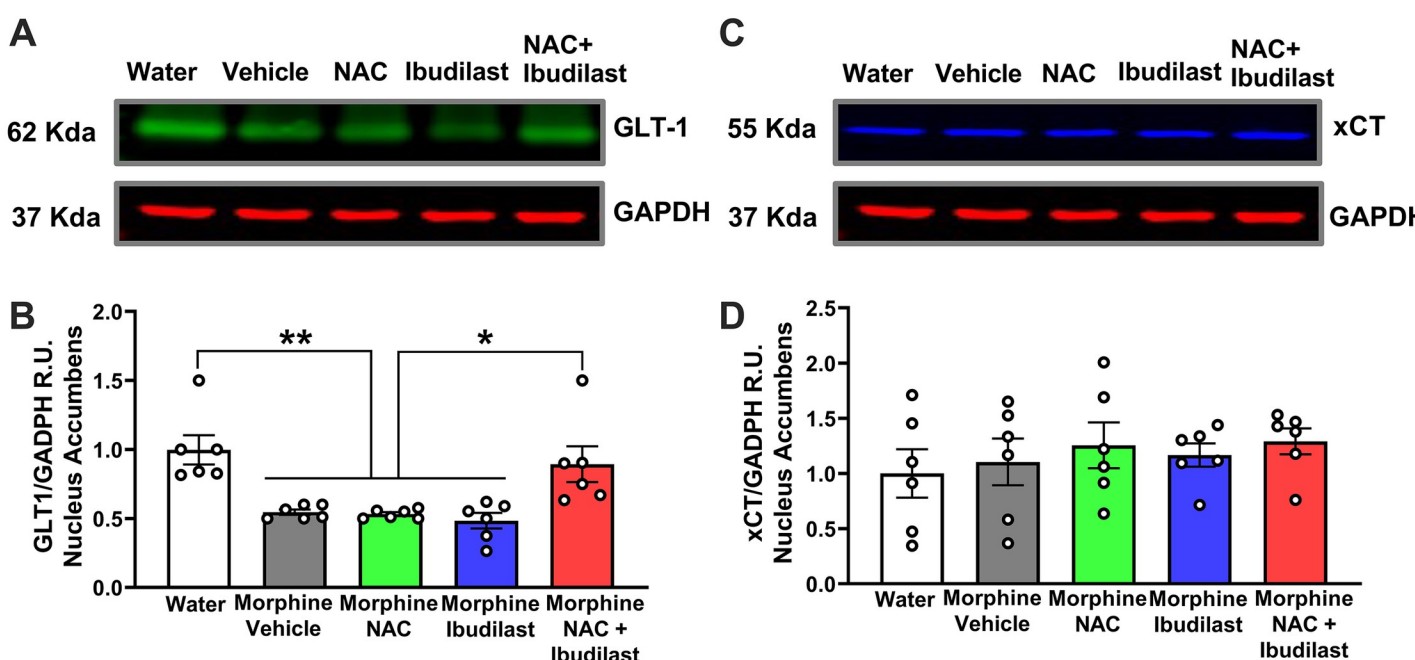

**Fig 6. The significant decrease in GLT-1 levels observed in the nucleus accumbens of rats that had consumed morphine for fifty-six days was completely reversed by the co-administration of NAC (40 mg/kg/day, i.p.) and ibudilast (7.5 mg/kg/day, i.p.).** Shapiro-Wilk tests conducted for both the groups of GLT-1 protein levels data (A) and the groups of xCT protein levels data (B) indicated that both datasets met the criteria for normality (alpha = 0.05), implying that the data of GLT-1 protein levels and that of xCT protein levels follow a normal distribution. **(A)** Representative Western Blot image showing GLT-1 detection in the nucleus accumbens, GAPDH was used as housekeeping marker. **(B)** Quantification of GLT-1 levels. Chronic morphine consumption induced a significative reduction in GLT-1 protein level in the nucleus accumbens of morphine drinking rats treated with vehicle, NAC, or ibudilast compared to rats that drank only water (**p<0.001, One-way ANOVA [$F_{\text{treatment}(4,25)}$ = 9.04, p<0.0001, followed by Tukey post-hoc test, n = 6 animals per group]. Co-administration of NAC + ibudilast completely restored the protein level of GLT-1 in the nucleus accumbens of morphine-drinking rats (*p<0.05) compared to vehicle, NAC, or ibudilast-treated morphine-drinking rats. **(C)** Representative Western Blot image showing xCT detection in the nucleus accumbens, GAPDH was used as housekeeping marker. **(D)** Quantification of xCT levels. Chronic morphine consumption or the administration of NAC, ibudilast or NAC + ibudilast did not affect xCT protein levels in the nucleus accumbens [$F_{\text{treatment}(4,25)}$ = 0.426, p: 0.78, N.S.].

shown to induce marked dependence, as evidenced significant alterations in an 8-sign naloxone-induced withdrawal score panel [36]. The sub-chronic intraperitoneal administration of NAC (over six days) or ibudilast (over four days) reduced morphine self-administration by 36% and 48% (p<0.0001) respectively. When NAC and ibudilast were administered together, morphine self-administration was reduced by 57% (p<0.001). The combined administration of NAC and ibudilast was significantly more effective than each treatment alone, specifically in (a) inhibiting morphine self-administration (Fig 1; p<0.001 and p<0.01), (b) normalizing astrocyte density (Fig 4; p<0.001 and p<0.01), and (c) normalizing glutamate GLT-1 transporter levels (Fig 6; p<0.05). We have previously demonstrated that there is an association between morphine-induced reactive astrocytes and demyelination in the prefrontal cortex in this rat model [56]. Thus, as a potential translational approach, the co-administration of ibudilast and NAC appears preferable to counteract chronic morphine effects. The reduction in morphine consumption in a female rat model of opioid dependence, induced by dual treatment with NAC plus ibudilast, could be particularly valuable for pregnant women with opioid use disorder, as it may allow for the administration of lower doses of methadone or buprenorphine to prevent withdrawal and reduce external drug-seeking behavior.

Previous studies showed that the administration of NAC alleviated acute methadone withdrawal behaviors in rats [57]. Furthermore, it has been demonstrated that ibudilast significantly reduced both spontaneous and naloxone-precipitated morphine withdrawal syndrome in rats [6]. Similarly, recent studies have indicated that NAC administration significantly reduced naloxone-precipitated fentanyl withdrawal syndrome [33]. These effects of NAC and ibudilast suggest that their combined administration is preferable and safer.

In previous studies [18, 36], we reported an increase in morphine-induced neuroinflammation and oxidative stress in nucleus accumbens, neostriatum and in the prefrontal cortex. In this study, we focused on the hippocampus, due its role as an integrative center for regulating the association between drug-related cues and drug rewards in opioid addiction, and due to the high expression of μ-opioid receptors on hippocampal neuronal and glial cells [58]. Current studies support findings that Wistar rats subjected to repeated systemic administration of morphine exhibit oxidative stress in the hippocampus, as evidenced by elevated levels of GSSG and MDA [1, 2]. Additionally, these findings align with research describing an association between morphine administration and microglial activation in the nucleus accumbens [59]. Morphological changes in glial cells represent one of the canonical characteristics of activated glia, serving as a hallmark of neuroinflammation [60, 61]. Our recent findings, which reveal a mechanistic link between reduced microglial branching and increased release of the inflammatory cytokine interleukin-1β (IL-1β) in methadone-exposed purified primary cultures of microglia and astrocytes, support the idea that opioids promote a pro-inflammatory state in both microglial and astrocytic cells, and highlight the role of glial activation in the neuroinflammatory response associated with opioid exposure [37]. Indeed, we have shown that methadone exposure in purified primary cultures of microglia and astrocytes induces a significant increase in the percentage of microglia expressing reactivity-associated markers (CD11b[high] and CD45[high]/Iba-1), accompanied by elevated mRNA levels of IL-6 and IL-8. Additionally, we observed an increase in TNF-α mRNA level in both microglial and astrocyte cultures and a rise in CCL5 mRNA in the astrocyte cell line, compared to the respective vehicle-treated cells [37].

The observation of proinflammatory morphologic glial activation, indicated by an elevated ratio of reactive to surveillant microglia, and a higher density of microglia and astrocytes in the CA1 region of the hippocampus, as accompanied by oxidative stress in the same brain region, as shown by an elevated ratio of GSSG/GSH and elevated MDA levels, underscores the correlation between inflammatory responses and oxidative stress in the context of chronic morphine intake. An opioid induced switch of microglia morphology toward a reactive state has been

recently reported by others [62]. In this case, authors reported that heroin self-administration induces microglia reactivity without changes in cell density in the medial prefrontal cortex but not in insula, suggesting a region-specific effect [62]. These results align with previous studies demonstrating that rats treated with morphine via subcutaneous or intrathecal routes for 7–9 days showed neuroinflammation, as measured by the release of proinflammatory cytokines [63] and by an increase in immunohistochemical markers of microglia and astrocytes activation in hippocampus [4, 6, 64]. Additionally, it has been reported that glial activation induces a transcriptomic reprogramming with up-regulation of pro-inflammatory and immune signaling pathways across multiple nodes in the addiction circuitry of the opioid-exposed brain, thereby pointing to heritable risk architectures in the genomic organization of the brain's reward circuitry [65].

Regarding the mechanisms leading to opioid self-administration, acute morphine administration exerts its rewarding effects by activating μ-opioid receptors on GABAergic neurons in the ventral tegmental area (VTA), which leads to the disinhibition of dopaminergic neurons that project to the nucleus accumbens [66]. However, with chronic morphine use, the glutamatergic input on neurons of the nucleus accumbens is enhanced [67], resulting in a long-lasting and progressive increase in the opioid rewarding effect. This increase is thought to play an important role in the maintenance of drug addiction [67]. Several preclinical studies have shown that synaptic glutamate overflow in the nucleus accumbens results from the downregulation of the glutamate transporter GLT-1 observed in heroin self-administering rats [21, 55, 68]. This downregulation becomes a critical mechanism underlying the reinstatement of drug seeking and relapse [21, 55, 68]. Consequently, the ability of astrocytes in the nucleus accumbens to reuptake extracellular glutamate is decreased, in animals consuming morphine.

A downregulation of GLT-1 levels was indeed observed in the morphine-drinking rats in the present study. This may be due to morphine-induced oxygen radicals that can oxidize cysteine residues of the GLT-1 protein, thereby inactivating it [69]. This inactivation can lead to an increased glutamatergic tone, triggered by Pavlovian cues (including circadian interoceptive), which in turn may initiate morphine-seeking behavior. The present study reveals a 50% reduction in GLT-1 levels in nucleus accumbens after chronic morphine intake, making it the first demonstration that GLT-1 is downregulated during chronic opioid self-administration, rather than just being downregulated upon the withdrawal from opioid self-administration [55].

We suggest that two mechanisms are involved in the combined effects of NAC and ibudilast in reducing opioid intake (similar to a reduction in craving): (a) the combination of both agents, likely through their anti-inflammatory effects on glial inflammation (Fig 4C) [70] leads to increased levels of the glutamate transporter GLT-1 (Fig 6A). Elevated GLT-1 levels are known to reduce drug craving for many addictive substances [23, 71]. This view is consistent with studies showing that ceftriaxone administration increases GLT-1 expression and inhibits the reinstatement of heroin-seeking behavior [55]. Additionally, (b) NAC, by increasing cystine levels has been shown to enhance cystine-glutamate exchange in astrocytes [72]. This leads to the activation of the inhibitory mGlu2/3 receptors, which play a negative modulatory role in the release of glutamate at the synaptic cleft [73–75]. In line with this, it has been demonstrated that morphine inhibits cystine transport [9], suggesting that, prior to NAC administration, low cystine levels may reduce xCT activity. This reduction could fail to activate the inhibitory mGlu2/3 receptors, thereby increasing glutamate overflow at the tripartite synapse [76–78].

In several therapeutic approaches utilizing NAC, a loading dose is often recommended, followed by lower maintenance doses [79]. This study demonstrates that sub-chronic treatment with NAC, involving a loading dose of 70 mg/kg/day for two days followed by a maintenance dose of 40 mg/kg for four days, significantly inhibit morphine intake.

A note is in order on the dose of ibudilast. Due to the faster clearance of ibudilast in rats compared to humans [80, 81], the intraperitoneal dose of ibudilast used in this study (7.5 mg/kg/day) exceeds the typical oral dose used in human for asthma treatment (10 mg two or three times a day). Notably, ibudilast's half-life is 19 hours in humans [80] but only 45 minutes in rats [81].

Overall, we have shown that the concurrent use of the antioxidant NAC and the anti-inflammatory ibudilast induces a 55 to 60% reduction in chronic opioid consumption. This finding has valuable translational implications for the treatment of opioid use disorders, especially in cases where the continuous administration of oral opioid agonists, such as methadone, is considered a viable therapeutic option.

## Supporting information

**S1 Fig. Rats that had been pretreated with a daily intraperitoneal dose of morphine (40 mg/kg) for nine days demonstrated a progressive increase in their voluntary consumption of oral morphine, while having free access to both water and a morphine solution with concentrations that were incrementally raised over time.** Figure shows morphine consumption expressed as mg of morphine consumed per kilogram of body weight per day (mean ± SEM) of rats given free choice between a morphine sulfate solution of increasing concentration and water 24 hours/day (n = 24). Arrows indicate intraperitoneal administration of morphine. Two-way ANOVA (concentration × day) of all voluntary oral morphine intake data indicated a significant effect of the concentration of the morphine solution offered to rats [$F_{concentration(10,253)}$ = 351.5, p<0.0001], day [$F_{day(1548,400.8)}$ = 11.61, p<0.0001] and concentration × day interaction [$F_{interaction(20,506)}$ = 4.731, p<0.0001]. Tukey's post hoc test revealed that each increase in morphine concentration in the range of 6 mg/L to 90 mg/L resulted in a significant increase (****p<0.0001) in daily morphine intake.
(TIF)

**S2 Fig. Experimental timeline.** The timeline outlines the following events performed on the rats: (i) Intraperitoneal administration of morphine; (ii) The option to choose between oral morphine and water; (iii) On day 51, rats voluntarily drinking morphine received intraperitoneal loading doses of NAC or saline; (iv) On day 53, rats continuing to drink morphine and received intraperitoneal maintenance doses of vehicle, NAC, Ibudilast, or NAC + ibudilast; (v) On day 57, brain tissue samples were collected.
(TIF)

**S3 Fig. Original uncropped images of cropped blots of Fig 6A and 6C.** The lanes of the unedited blots that appear in the cropped images in the manuscript are highlighted.
(TIF)

## Acknowledgments

The technical assistance of Robel Vazquez, Juan Santibañez and Carmen Almeyda is greatly appreciated.

## Author Contributions

**Conceptualization:** María Elena Quintanilla, Paola Morales, Yedy Israel, Fernando Ezquer.

**Data curation:** María Elena Quintanilla, Paola Morales, Daniela Santapau, Javiera Gallardo, Rocío Rebolledo, Gabriel Riveras, Tirso Acuña, Fernando Ezquer.

**Formal analysis:** María Elena Quintanilla, Paola Morales, Daniela Santapau, Javiera Gallardo, Rocío Rebolledo, Gabriel Riveras, Tirso Acuña, Yedy Israel, Fernando Ezquer.

**Funding acquisition:** Mario Herrera-Marschitz, Fernando Ezquer.

**Investigation:** María Elena Quintanilla, Paola Morales, Javiera Gallardo, Mario Herrera-Marschitz, Fernando Ezquer.

**Methodology:** María Elena Quintanilla, Paola Morales, Daniela Santapau, Javiera Gallardo, Rocío Rebolledo, Gabriel Riveras, Tirso Acuña.

**Project administration:** María Elena Quintanilla.

**Supervision:** María Elena Quintanilla, Paola Morales.

**Writing – original draft:** María Elena Quintanilla, Yedy Israel, Fernando Ezquer.

**Writing – review & editing:** María Elena Quintanilla, Paola Morales, Daniela Santapau, Javiera Gallardo, Rocío Rebolledo, Gabriel Riveras, Tirso Acuña, Mario Herrera-Marschitz, Yedy Israel, Fernando Ezquer.

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
