## [Decision Letter · Decision Letter 0]

23 Jul 2024

PONE-D-24-20382Morphine self-administration is inhibited by the antioxidant N‐acetylcysteine and the anti-inflammatory Ibudilast; an effect enhanced by their co-administration.PLOS ONE

Dear Dr. Ezquer,

Thank you for submitting your manuscript to PLOS ONE. After careful consideration, we feel that it has merit but does not fully meet PLOS ONE’s publication criteria as it currently stands. Therefore, we invite you to submit a revised version of the manuscript that addresses the points raised during the review process.

Please address the comments raised by both reviewers.

We look forward to receiving your revised manuscript.

Kind regards,

Shao-Jun Tang

Academic Editor

PLOS ONE

Journal Requirements:

2. Thank you for stating the following financial disclosure: "This work was supported by Agencia Nacional de Investigación y Desarrollo (ANID) FONDECYT 1240162 and ACT210012 grants to Fernando Ezquer, and FONDECYT 1231443 to Mario Herrera-Marschitz."

3. Thank you for stating the following in the Acknowledgments Section of your manuscript: "This work was supported by Agencia Nacional de Investigación y Desarrollo (ANID) FONDECYT 1240162 and ACT210012 grants to Fernando Ezquer, and FONDECYT 1231443 to Mario Herrera-Marschitz. The technical assistance of Robel Vazquez, Juan Santibañez and Carmen Almeyda is greatly appreciated".

Please remove any funding-related text from the manuscript and let us know how you would like to update your Funding Statement. Currently, your Funding Statement reads as follows: "This work was supported by Agencia Nacional de Investigación y Desarrollo (ANID) FONDECYT 1240162 and ACT210012 grants to Fernando Ezquer, and FONDECYT 1231443 to Mario Herrera-Marschitz."

4. We note that your Data Availability Statement is currently as follows: "All relevant data are within the manuscript and its Supporting Information files."

Reviewers' comments:

Reviewer's Responses to Questions

**Comments to the Author**

1. Is the manuscript technically sound, and do the data support the conclusions?

Reviewer #1: Yes

Reviewer #2: Yes

2. Has the statistical analysis been performed appropriately and rigorously? 

Reviewer #1: Yes

Reviewer #2: Yes

3. Have the authors made all data underlying the findings in their manuscript fully available?

Reviewer #1: Yes

Reviewer #2: Yes

4. Is the manuscript presented in an intelligible fashion and written in standard English?

Reviewer #1: Yes

Reviewer #2: Yes

5. Review Comments to the Author

Reviewer #1: The research done by the authors is well planned and justified, and the results are clearly described and showed in the figures. The paper itself is very interesting and provides the readers with new information regarding the dependence and new possibilities in lowering the adverse effects of opioid withdrawal and decreasing the level of dependence in using easy, safe and affordable measures. My suggestion to the authors is to reorganize or add extra information in the discussion section regarding the limitation of the study (example - M1/M2, ibudilast clearance) and a conclusions part in the end of the discussion to summarize the study, and give the reader an answer to the question stated by the authors while planning the study - yes, NAC might be helpful, but also, shortly, by means of what mechanisms/processes? In my opinion this will make the discussion less results-like and more pleasant to read and understandable.

Reviewer #2: Quintanilla and colleagues tested the effect of NAC, ibudilast and their combination on rats subjected to prolonged morphine self-administration in a TBC paradigm. Both drugs reduced morphine intake and the combination showed further efficacy. Then they they demonstrated that NAC, ibudilast and their combination reverted the effect of morphine on

biomarkers of oxidative stress and glial activation in the hippocampus. They also reported that morphine decreased the level of GLT1 in the accumbens, and demonstrated that the combination of NAC and ibudilast, but not the single drugs, could revert this effect of morphine.

The study is overall interesting and soundly built. There are a number of points that however, require further attention.

Authors should better rationalize the choice for working in female only. The fact that male rats consume less morphine than females (lines 162-163) does not mean they don’t consume morphine or that males are not subject to opioid addiction risks. If there is not a stronger rational, this should be recognized as a study limitation.

Morphine has a low oral bioavailability. Although oral morphine is used in pain therapy, the oral way is not the route normally chosen by people affected by OUD. To which extent the oral way is used to abuse morphine? Also, when a patient assuming morphine for treatment purposes develops OUD, they often switch to other opioids cheaply available on the black market, like heroin.

Lines 208-209, please move consumption data to the result section.

Supplementary Figure 1 show two factors, the morphine concentrations and the day each concentration was maintained, therefore a two-way anova would be a better choice here. However, it looks like not all concentration were maintained the same number of days, if that’s the case authors should consider using the average intake of each concentration and keep the one-way ANOVA.

Experiment in figure 1: Baseline training; please provide statistic. NAC loading phase lasted two days, why using a one-way anova? NAC + Ibudilast phase, please clarify to which day the significant difference between groups refer to.

Same for figure 2.

In TBC paradigm food intake is used as natural reinforcer control, did you check food intake during treatments?

Please provide the rational for the euthanasia time point. Were rats still under morphine when sacrificed?

Result section usually report exclusively results. In this regard, what is written in

Line 348-352 and 366-367 look better suited for sections like introduction or discussion.

Line 957 the df do not correspond to what a one way ANOVA with 5 groups of 6 rats each would yield.

Line 932 please check df.

The classical view of dichotomous microglia M1/M2 states has been challenged as too simplistic to describe the multidimensional activation profiles of microglia. Authors may want to have a look at PMID: 36327895 and if they found the arguments in there reasonable, they could reconsider the terminology used in this work.

Also, opioid induced switch of microglia morphology toward a reactive state has been recently reported by others (PMID: 38452987). In this case authors reported that opioids affected microglia morphology but not density.

There are typos scattered over the manuscript please correct.

6. PLOS authors have the option to publish the peer review history of their article (what does this mean?). If published, this will include your full peer review and any attached files.

Reviewer #1: No

Reviewer #2: No

---

## [Author Response · Author response to Decision Letter 0]

7 Aug 2024

August 8th, 2024

Dr. Shao-Jun Tang

Academic Editor

PLOS ONE

Dear Dr. Tang: 

Re: ID PRONE-D-24-20382

Thank you for inviting us to revise our manuscript entitled “Morphine self-administration is inhibited by the antioxidant N-acetylcysteine and the anti-inflammatory ibudilast; an effect enhanced by their co-administration” by María Elena Quintanilla, Paola Morales, Daniela Santapau, Javiera Gallardo, Rocio Rebolledo, Gabriel Rivas, Tirso Acuña, Mario Herrera-Marschitz, Yedy Israel and myself. We are pleased to learn that the reviewers found our work to be of interest and we are grateful for their constructive comments toward the improvement of the manuscript.

We have acted on all the comments received and have modified our manuscript accordingly. Please find below a point-by-point reply organized as Reviewer’s Comment; Reply and Modifications. The latter are incorporated in the resubmitted manuscript (clean copy). Modifications on the revised manuscript are highlighted in red (marked-up copy). 

We hope that the recommended changes incorporated on our manuscript will make our revised version acceptable for publication in PLOS ONE.

Sincerely yours, 

Fernando Ezquer, PhD.

Center for Regenerative Medicine

School of Medicine, Universidad del Desarrollo

Santiago, CHILE

eezquer@udd.cl

POINT-BY-POINT REPLIES AND MODIFICATIONS.

Editor

Editor-Comment 1: Thank you for stating the following financial disclosure: "This work was supported by Agencia Nacional de Investigación y Desarrollo (ANID) FONDECYT 1240162 and ACT210012 grants to Fernando Ezquer, and FONDECYT 1231443 to Mario Herrera-Marschitz. Please state what role the funders took in the study. If the funders had no role, please state: "The funders had no role in study design, data collection and analysis, decision to publish, or preparation of the manuscript.

Editor Reply and Modification 1: Many thanks for your comment. In the financial disclosure section and in the cover letter we added the following sentence: The funders had no role in study design, data collection and analysis, decision to publish, or preparation of the manuscript. 

Editor-Comment 2: Thank you for stating the following in the Acknowledgments Section of your manuscript: "This work was supported by Agencia Nacional de Investigación y Desarrollo (ANID) FONDECYT 1240162 and ACT210012 grants to Fernando Ezquer, and FONDECYT 1231443 to Mario Herrera-Marschitz. The technical assistance of Robel Vazquez, Juan Santibañez and Carmen Almeyda is greatly appreciated".

We note that you have provided funding information that is not currently declared in your Funding Statement. However, funding information should not appear in the Acknowledgments section or other areas of your manuscript. We will only publish funding information present in the Funding Statement section of the online submission form. Please remove any funding-related text from the manuscript and let us know how you would like to update your Funding Statement. Currently, your Funding Statement reads as follows: "This work was supported by Agencia Nacional de Investigación y Desarrollo (ANID) FONDECYT 1240162 and ACT210012 grants to Fernando Ezquer, and FONDECYT 1231443 to Mario Herrera-Marschitz."

Editor Reply and Modification 2: Thank you for this comment. In the resubmitted version of our manuscript, we removed the founding information from the Acknowledgments section. 

Editor-Comment 3: We note that your Data Availability Statement is currently as follows: "All relevant data are within the manuscript and its Supporting Information files. Please confirm at this time whether or not your submission contains all raw data required to replicate the results of your study. Authors must share the “minimal data set” for their submission. PLOS defines the minimal data set to consist of the data required to replicate all study findings reported in the article, as well as related metadata and methods.

Editor Reply and Modification 3: We confirm that our submission contains all raw data required to replicate the results of the study.

Editor-Comment 4: PLOS ONE now requires that authors provide the original uncropped and unadjusted images underlying all blot or gel results reported in a submission’s figures or Supporting Information files. When you submit your revised manuscript, please ensure that your figures adhere fully to these guidelines and provide the original underlying images for all blot or gel data reported in your submission. 

In your cover letter, please note whether your blot/gel image data are in Supporting Information or posted at a public data repository, provide the repository URL if relevant, and provide specific details as to which raw blot/gel images, if any, are not available. 

Editor Reply and Modification 4: Many thanks for your comment. Original uncropped images of cropped blots of Figure 6A and 6C were incorporated in the Supplemental Material as Supplemental File 3 in the resubmitted version of our manuscript. The lanes of the unedited blots that appear in the cropped images in the manuscript are highlighted.

Editor-Comment 5: We note that you have included the phrase “data not shown” in your manuscript. Unfortunately, this does not meet our data sharing requirements. PLOS does not permit references to inaccessible data. We require that authors provide all relevant data within the paper, Supporting Information files, or in an acceptable, public repository. Please add a citation to support this phrase or upload the data that corresponds with these findings to a stable repository (such as Figshare or Dryad) and provide and URLs, DOIs, or accession numbers that may be used to access these data. Or, if the data are not a core part of the research being presented in your study, we ask that you remove the phrase that refers to these data.

Editor Reply and Modification 5: Thank you very much for your comment. Following Editor suggestion in the resubmitted version of our manuscript we have deleted the phrase that refers to data not shown.

Editor-Comment 6: Please include captions for your Supporting Information files at the end of your manuscript, and update any in-text citations to match accordingly.

Editor Reply and Modification 6: Thank you for your comment. This recommendation was implemented at the end of our manuscript.

Editor-Comment 7: Please review your reference list to ensure that it is complete and correct. If you have cited papers that have been retracted, please include the rationale for doing so in the manuscript text, or remove these references and replace them with relevant current references. Any changes to the reference list should be mentioned in the rebuttal letter that accompanies your revised manuscript.

Editor Reply and Modification 7: We reviewed the reference list and we confirm that it is complete and correct.

Reviewer #1

Rev#1-Comment 1: The research done by the authors is well planned and justified, and the results are clearly described and showed in the figures. The paper itself is very interesting and provides the readers with new information regarding the dependence and new possibilities in lowering the adverse effects of opioid withdrawal and decreasing the level of dependence in using easy, safe and affordable measures. 

Rev#1 Reply and Modification 1: Many thanks for your kind comment. No modification needed. 

Rev#1-Comment 2: My suggestion to the authors is to reorganize or add extra information in the discussion section regarding the limitation of the study (example - M1/M2, ibudilast clearance) and a conclusions part in the end of the discussion to summarize the study, and give the reader an answer to the question stated by the authors while planning the study - yes, NAC might be helpful, but also, shortly, by means of what mechanisms/processes? In my opinion this will make the discussion less results-like and more pleasant to read and understandable.

Rev#1 Reply and Modification 2: Many thanks; we agree. There are three major changes introduced in the modified manuscript: 

(i) While conducting gene expression analyses of opioid-induced glial changes remains a preferred method to evaluate neuroinflammation, Reviewer 2 provided new recommendations and references [PMID: 36327895 and PMID: 38452987], which we have incorporated. Based on these recommendations, we now use the ratio of activated to surveillance glial cells as an indication of glial activation and neuroinflammation throughout the manuscript.

(ii) We have deleted the human vs rat comparison of ibudilast doses in the "Conclusions" section as these do not alter the conclusions or translational extrapolation. 

(iii) We have explained the mechanism by which NAC could contribute to reducing drug craving of addictive drugs via activating the xCT-mediated cystine-glutamate exchange leading to the activation of the inhibitory Glu2/3 receptor, and have included supporting references in the Discussion section.

Reviewer #2

Rev#2-Comment 1: Quintanilla and colleagues tested the effect of NAC, ibudilast and their combination on rats subjected to prolonged morphine self-administration in a TBC paradigm. Both drugs reduced morphine intake and the combination showed further efficacy. Then they they demonstrated that NAC, ibudilast and their combination reverted the effect of morphine on biomarkers of oxidative stress and glial activation in the hippocampus. They also reported that morphine decreased the level of GLT1 in the accumbens, and demonstrated that the combination of NAC and ibudilast, but not the single drugs, could revert this effect of morphine.The study is overall interesting and soundly built. 

Rev#2 Reply and Modification 1: Many thanks for your kind comment. No modification needed. 

Rev#2-Comment 2: Authors should better rationalize the choice for working in female only. The fact that male rats consume less morphine than females (lines 162-163) does not mean they don’t consume morphine or that males are not subject to opioid addiction risks. If there is not a stronger rational, this should be recognized as a study limitation.

Rev#2 Reply and Modification 2: Thank you very much. We agree that male rats can also consume morphine and develop an addiction to it. In the revised version of the manuscript (Materials and Methods section, lines 159 ton 166), we now indicate: "The rational for choosing to work only with females was twofold: (i) we had previously shown in our laboratory that female rats of this line developed morphine dependence after reaching a stable morphine intake, as in the present study (15.6 ± 0.8 mg/kg/day), using the same morphine intake induction method (Quintanilla et al 2023, doi: 10.3390/ijms242317081); and, (ii) female rats were used for translational considerations, such as the acute need to tone down the opioid dependence in pregnant women who are normally treated with long-acting opioids (e.g methadone). We have recently shown that methadone is per se a neurotoxin (De Gregorio et al 2024, doi: 10.1038/s41598-024-67860-7). The fact that we did not include males in our study is a limitation that we now indicated. 

Rev#2-Comment 3: Morphine has a low oral bioavailability. Although oral morphine is used in pain therapy, the oral way is not the route normally chosen by people affected by OUD. To which extent the oral way is used to abuse morphine? Also, when a patient assuming morphine for treatment purposes develops OUD, they often switch to other opioids cheaply available on the black market, like heroin.

Rev#2 Reply and Modification 3: Thank you very much for this comment. Most often, animal models only replicate one relevant aspect, while not all the characteristics seen in humans. Opioid dependence is the main characteristic of this model - In a previous study conducted in our laboratory opioid dependence (assessed by naloxone-precipitated withdrawal syndrome) was demonstrated (Quintanilla et al., 2023, doi: 10.3390/ijms242317081).

Regarding the extent to which morphine is abused orally, we note that in humans over 90% of prescription opioid abusers report oral ingestion for nonmedical purposes (McCabe et al., 2007, doi: 10.1016/j.addbeh.2006.05.022; Crummy et al., 2020, doi: 10.3389/fnins.2020.00569). Additionally, the morphine used in this study is specifically formulated for oral administration (morphine sulfate, Oramorph), increasing its bioavailability after oral administration. This information has been added to the Material and Methods section, lines 193 to 197.

Rev#2-Comment 4: Lines 208-209, please move consumption data to the result section.

Rev#2 Reply and Modification 4: Many thanks for this comment. Following reviewer’s suggestion, the consumption data were moved to the Result section lines 310 to 313.

Rev#2-Comment 5: Supplementary Figure 1 show two factors, the morphine concentrations and the day each concentration was maintained, therefore a two-way anova would be a better choice here. However, it looks like not all concentration were maintained the same number of days, if that’s the case authors should consider using the average intake of each concentration and keep the one-way ANOVA.

Rev#2 Reply and Modification 5: Many thanks for this comment. Since all concentration differences were maintained for 3 days, data in Supplementary Figure 1 were now analyzed using a two-way analysis of variance (ANOVA) followed by Tukey's post hoc test. We have included this change in both the marked copy of the revised manuscript (Result section, line 314) and in the marked copy of the Supplementary Material (legend to supplementary figure 1).

Rev#2-Comment 6: Experiment in figure 1: Baseline training; please provide statistic. NAC loading phase lasted two days, why using a one-way anova? NAC + Ibudilast phase, please clarify to which day the significant difference between groups refer to. Same for figure 2.

Rev#2 Reply and Modification 6: Many thanks for this comment. The resubmitted version of the manuscript has been improved in all aspects suggested by the reviewer. This reply addresses comments for both Figure 1 and 2 as follows:

(a) Baseline consumption statistics.

 Figure 1: The two-way ANOVA of morphine intake during the initial training of the four groups is now mentioned in the Results section (line 314), while the complete statistical analysis is presented in the legend of Figure 1 (lines 955 to 957).

 Figure 2A: Two-way ANOVA of the water intake during the baseline training of the four groups has been included in the Results section (lines 338 to 339) and in the legend corresponding to Figure 2A (lines 984 to 986).

 (b) NAC loading phase. Now, a two-way analysis of variance was performed. 

 Figure 1: A two-way ANOVA (treatment × day) from all four groups on days 50 and 51 of treatment during the administration of a loading dose of NAC (70 mg/kg/day) to rats in both the NAC and NAC + ibudilast groups revealed a significant effect of the loading dose of NAC in both groups compared to the other two vehicle-treated groups [Ftreatment(3,20) = 27.14, p<0.0001], but not of the day. Dunnett's post-hoc test indicated that the administration of NAC induced a significant reduction of morphine intake in both the NAC (***p< 0.001) and the NAC + ibudilast group (**p<0.01) compared to the groups treated with vehicle. Correction of ANOVA was included in the Results section, lines 315 to 321, and in the legend to Figure 1, lines 957 to 963.

 Figure 2A: A two-way ANOVA (treatment × day) was performed on water intake on both days (day 50 and day 51) of treatment with a NAC loading dose of 70 mg/kg/day in rats from both the NAC and NAC + ibudilast groups. The two-way ANOVA, followed by Dunnett's post hoc test, revealed a significant effect of the NAC loading dose compared to vehicle-treated groups [Ftreatment(3,20) = 27.14, p<0.0001], but not of the day. The statistical significance obtained is shown in the newly submitted Figure 2A, and the ANOVA correction has been included in the Results section (lines 339 to 342) and in the legend to Figure 2A (lines 986 to 993).

(c) Days chosen to analyze differences between groups: 

 Figure 1: The morphine intake data shown on days 53, 54, 55, and 56 by the four groups were chosen to analy

---

## [Decision Letter · Decision Letter 1]

2 Oct 2024

Morphine self-administration is inhibited by the antioxidant N‐acetylcysteine and the anti-inflammatory ibudilast; an effect enhanced by their co-administration.

PONE-D-24-20382R1

Dear Dr. Ezquer,

We’re pleased to inform you that your manuscript has been judged scientifically suitable for publication after fixing the minor comments from reviewer 2. 

Please address the comments and return the revised manuscript by Nov 16 2024 11:59PM.

Kind regards,

Shao-Jun Tang

Academic Editor

PLOS ONE

Reviewers' comments:

Reviewer's Responses to Questions

**Comments to the Author**

1. If the authors have adequately addressed your comments raised in a previous round of review and you feel that this manuscript is now acceptable for publication, you may indicate that here to bypass the “Comments to the Author” section, enter your conflict of interest statement in the “Confidential to Editor” section, and submit your "Accept" recommendation.

Reviewer #1: All comments have been addressed

Reviewer #2: All comments have been addressed

2. Is the manuscript technically sound, and do the data support the conclusions?

Reviewer #1: Yes

Reviewer #2: Yes

3. Has the statistical analysis been performed appropriately and rigorously? 

Reviewer #1: Yes

Reviewer #2: Yes

4. Have the authors made all data underlying the findings in their manuscript fully available?

Reviewer #1: Yes

Reviewer #2: No

5. Is the manuscript presented in an intelligible fashion and written in standard English?

Reviewer #1: Yes

Reviewer #2: Yes

6. Review Comments to the Author

Reviewer #1: Thank you for providing the revised manuscript. The authors have made all of the revisions suggested by the reviewers. I recommend the paper for publication.

Reviewer #2: Authors appropriately addressed my major points. I have only few remaining minor comments to make.

Line 230: Authors specified that animals were euthanized one hour after the last recorded morphine intake, it remains unclear however whether morphine was withdrawn during that hour (i.e. rats had one hour of abstinence) or not. This is a crucial information for the reproducibility of the study and I warmly recommend to clarify it.

Line 314 and Fig 1: when ANOVA finds no significant effects, it does not require to be followed by post-hoc tests. This apply also to other session of the manuscript were a non-significant effect is reported.

Both factors and their interaction of two-way ANOVA should be reported, whether they are significant or not.

7. PLOS authors have the option to publish the peer review history of their article (what does this mean?). If published, this will include your full peer review and any attached files.

Reviewer #1: No

Reviewer #2: No

---

## [Author Response · Author response to Decision Letter 1]

5 Oct 2024

October 5th, 2024

Dr. Shao-Jun Tang

Academic Editor

PLOS ONE

Dear Dr. Tang:

Re: ID PONE-D-24-20382R1: Final Decision Being Processed -

Thank you for informing us that our manuscript entitled “Morphine self-administration is inhibited by the antioxidant N-acetylcysteine and the anti-inflammatory Ibudilast; an effect enhanced by their coadministration” by María Elena Quintanilla, Paola Morales, Daniela Santapau, Javiera Gallardo, Rocio Rebolledo, Gabriel Rivas, Tirso Acuña, Mario Herrera-Marschitz, Yedy Israel and Fernando Ezquer has been judged scientifically suitable for publication in PLOS ONE, following the correction of minor comments by Reviewer 2.

We have addressed the minor comments provided by reviewer 2 and have modified our manuscript accordingly. Please find below a reply organized as Reviewer’s Comment; Reply and Modifications. The modifications have been incorporated into the resubmitted manuscript (clean copy), and the changes are highlighted in red in the marked-up copy.

We hope that the recommended changes incorporated on our manuscript will make this revised version acceptable for publication in PLOS ONE.

Sincerely, 

Fernando Ezquer, PhD

Center for Regenerative Medicine

School of Medicine

Universidad del Desarrollo

Chile

POINT-BY-POINT REPLIES AND MODIFICATIONS

Reviewer #2

Rev#2-Comment 1: Authors appropriately addressed my major points. I have only few remaining minor comments to make. Line 230: Authors specified that animals were euthanized one hour after the last recorded morphine intake, it remains unclear however whether morphine was withdrawn during that hour (i.e. rats had one hour of abstinence) or not. This is a crucial information for the reproducibility of the study, and I warmly recommend to clarify it.

Rev#2 Reply and Modification 1: Thank you very much for this important comment. Now we clarify (line 230): "Seventeen hours after the last administration of NAC and ibudilast, and one hour after the last recorded morphine intake, rats that continued to have access to the morphine solution and water were anesthetized with a cocktail of 60 mg/kg ketamine HCl, 10 mg/kg xylazine HCl and 4 mg/kg acepromazine (administered intramuscularly in a volume of 1.9 mL/kg) [43], intracardially perfused with 100 mL of 0.1 M PBS (pH 7.4) and euthanized”.

Rev#2-Comment 2: Line 314 and Fig 1: when ANOVA finds no significant effects, it does not require to be followed by post-hoc tests. This apply also to other sessions of the manuscript where a non-significant effect is reported.

Rev#2 Reply and Modification 2. Thank you very much for your comment. We have now removed mentions of post-hoc tests in the following locations of the Results section: lines 314, 349, and 351, as the ANOVAs did not find any significant differences.

---

## [Editor Report · Decision Letter 2]

15 Oct 2024

Morphine self-administration is inhibited by the antioxidant N‐acetylcysteine and the anti-inflammatory ibudilast; an effect enhanced by their co-administration.

PONE-D-24-20382R2

Dear Dr. Ezquer,

We’re pleased to inform you that your manuscript has been judged scientifically suitable for publication and will be formally accepted for publication once it meets all outstanding technical requirements.

Kind regards,

Shao-Jun Tang

Academic Editor

PLOS ONE
---

## [Editor Report · Acceptance letter]

18 Oct 2024

PONE-D-24-20382R2 

PLOS ONE

Dear Dr. Ezquer, 

I'm pleased to inform you that your manuscript has been deemed suitable for publication in PLOS ONE. Congratulations! Your manuscript is now being handed over to our production team.

Kind regards, 

on behalf of

Dr. Shao-Jun Tang 

Academic Editor

PLOS ONE